# Two Operational Modes of Cardio-Respiratory Coupling Revealed by Pulse-Respiration Quotient

**DOI:** 10.3390/bioengineering10020180

**Published:** 2023-01-31

**Authors:** Aleksandar Kalauzi, Zoran Matić, Mirjana M. Platiša, Tijana Bojić

**Affiliations:** 1Department for Life Sciences, Institute for Multidisciplinary Research, University of Belgrade, 11030 Belgrade, Serbia; 2Biomedical Engineering and Technologies, University of Belgrade, 11000 Belgrade, Serbia; 3Institute of Biophysics, Faculty of Medicine, University of Belgrade, P.O. Box 22, 11129 Belgrade, Serbia; 4Department of Radiation Chemistry and Physics 030, “VINČA” Institute of Nuclear Sciences—National Institute of the Republic of Serbia, University of Belgrade, P.O. Box 522, Mike Petrovića Alasa 12–14, 11000 Belgrade, Serbia

**Keywords:** heart rate variability, pulse respiration quotient, slow breathing, cardio-respiratory coupling, cardio-respiratory synchronization, sympatho-vagal balance

## Abstract

Due to the fact that respiratory breath-to-breath and cardiac intervals between two successive R peaks (*BBI* and *RRI*, respectively) are not temporally concurrent, in a previous paper, we proposed a method to calculate both the integer and non-integer parts of the pulse respiration quotient (*PRQ* = *BBI*/*RRI* = *PRQ_int_* + *b1* + *b2*), *b1* and *b2* being parts of the border *RRIs* for each *BBI*. In this work, we study the correlations between *BBI* and *PRQ*, as well as those between *BBI* and mean *RRI* within each *BBI* (*mRRI*), on a group of twenty subjects in four conditions: in supine and standing positions, in combination with spontaneous and slow breathing. Results show that the *BBI* vs. *PRQ* correlations are positive; whereas the breathing regime had little or no effect on the linear regression slopes, body posture did. Two types of scatter plots were obtained with the *BBI* vs. *mRRI* correlations: one showed points aggregated around the concurrent *PRQ_int_* lines, while the other showed randomly distributed points. Five out of six of the proposed aggregation measures confirmed the existence of these two cardio-respiratory coupling regimes. We also used *b1* to study the positions of R pulses relative to the respiration onsets and showed that they were more synchronous with sympathetic activation. Overall, this method should be used in different pathological states.

## 1. Introduction

One of the key properties of physiological rhythms (oscillations) is their ability (or tendency) to synchronize with each other. The general definition of synchronization is applicable to physiological rhythms and characterizes them in a very profound way: “synchronization is not a state, but a process of adjustment of rhythms due to interaction” [1]. Representing two physiological rhythms that are controlled by different but partly overlapping mechanisms, heart rhythm (the autonomic nervous system (ANS)) and breathing (partly automatic, partly volitional), coupling the two, as expressed through temporal synchronization, has attracted widespread research interest [1,2,3,4,5,6,7,8]. This is mainly related to questions as to whether and how their instantaneous periods are correlated in the time domain. However, a basic analytic difficulty, contained in the fact that their frequencies are not commeasurable and their instantaneous phases (e.g., ECG R peak positions and inspiration onsets) are not synchronized, has hampered this research goal. One very helpful approach to this problem could be a recently demonstrated methodological progress for calculating a particular aspect of cardio-respiratory coupling (CRC), named the pulse respiration quotient (*PRQ*, [9,10,11,12,13,14,15]), which has emerged as an extremely simple and useful technique, although representing an “untapped parameter” [11]. Briefly, *PRQ* is a variable resulting from counting the number of RR intervals (*RRI*) that corresponds to each breath-to-breath interval (*BBI*). In other words, it is a ratio between heart rate and breathing rate, or *BBI* and *RRI*: *PRQ* = *HR*/*BR* = *BBI*/*RRI*.

In our previous paper [16], we introduced an advanced method of *PRQ* calculation that comprised the precise determination of whole (integer)-number *RRI* values and the non-integer parts of two “border” RR intervals within each *BBI*. This approach also allowed for a more precise calculation of both integer and non-integer *PRQ* variability [16]. In addition, our analysis indicated that all *PRQ* values, measured from a group of twenty healthy human individuals, recorded in four experimental conditions [17], tended to aggregate around integers—lower values were recorded in conditions with spontaneous breathing (as expected, see [9,10,11]), but higher integer values were also recorded in conditions with slow 0.1 Hz breathing. The latter result was, to a certain extent, unexpected, demonstrating that this “integer affinity” extended to a non-spontaneous breathing regime, such as with externally imposed slow breathing. Moreover, by dividing each *BBI* with this new, more precise *PRQ* value, we were able, for the first time, to introduce a new operational variable—mean *RRI* (*mRRI*), which was valid for each particular *BBI* (mean *RRI* per *BBI*). This allowed us to obtain two directly comparable and temporally synchronized data series, each generated by one of the two cardio-respiratory sub-systems. The study of their correlation, in terms of time domain, became easier than with some previously used techniques [2,11,18]. In addition, by studying the scatter plots of such obtained respiratory and mean heart intervals (*BBI* and *mRRI*), we try to explain in some detail the properties of their correlation, and possibly reveal the mechanism behind this physiological integer *PRQ* preference over a mere random *PRQ* distribution, although both types were detected in our experiments. Therefore, this fundamental and still not completely answered question about the “quantification of [cardio-respiratory] synchro-grams” posed by Schäfer et al. [2], reformulated (transferred) into the question of *PRQ* quantization, was our research preoccupation in this work. To be more specific, we tried to give a physical and physiological explanation for *PRQ* quantization and to indicate its physiological significance in healthy cardio-respiratory function. Owing to the aforementioned computational advancements, it seems that the quantization itself became more accessible and easier to explain via the concept of *PRQ* than by means of earlier synchro-gram analysis. Basically, our approach to studying the phenomenon of *PRQ* quantization was begun by: (i) an analysis of the correlation between *BBI* and *PRQ* that was initially suggested by Scholkmann and Wolf [11], but was just in the form of a plot, without detailed characterization of this relationship, and (ii) the calculation of mean *RRI* for each *BBI*. We characterized these relationships, and especially their patterns of aggregation vs. randomness, by means of several indices. It turned out that these indices could be used to discriminate the experimental conditions with sympathetic domination from those with vagal domination, which was not possible earlier with the use of classical linear measures.

## 2. Materials and Methods

In this research, we used experimental data from our previous study, described in detail in Ref. [17]. For the purposes of a clear methodological framework, in this work, we are going to give only a brief overview.

### 2.1. Subjects

This study included 20 healthy adult human subjects with a mean age of 34.4 ± 7.4, of whom 13 were males. Subjects were selected using the appropriate health standards and criteria of inclusion and exclusion; this research with human subjects was approved by the Ethical Committee of the Faculty of Medicine, University of Belgrade (No. 2650/IV-24).

### 2.2. Study Protocol

The ECG and breathing signals of 20 subjects were recorded in four physiological states: in a supine position with spontaneous breathing (supin); while standing, with spontaneous breathing (stand); a supine position with 0.1 Hz breathing (supin01); standing, with 0.1 Hz breathing (stand01). The recordings of signals were made in the same period of the day for all subjects (between 8 and 12 a.m.) in order to have the same circadian circumstances regarding the autonomic control of physiological rhythms. Recording in each of the four physiological states lasted for twenty minutes, with a five-minute break between them, inserted for the purpose of autonomic stabilization after body posture transition (from supine to standing and vice versa). These experimental procedures were performed under controlled laboratory conditions. Subjects were introduced to and prepared for the experimental procedures before each recording, with special attention paid to correctly performing slow breathing at 0.1 Hz frequency, by gradually adjusting to breathing in five-second-long inspirations and expirations that were dictated by a metronome sound.

### 2.3. Data Acquisition

For ECGs and breathing-signal acquisition, a Biopac MP100 system (Biopac System, Inc., Santa Barbara, CA, USA; AcqKnowledge 3.91 software) was used. We registered the ECG signal via the attachment of ECG lead electrodes on the projections of clavicle bones and the ankle of the right leg for grounding. For the continuous recording of the breathing signal, we used a belt with a resistive strain gauge transducer positioned about a centimeter above the costal line. The sampling of ECG and breathing signals was performed with the same frequency of 1000 Hz. Filters were set according to the instructions given by Biopac for general measurement usage: the gain was set to 10 and low-pass filtering to 10 Hz, with no high-pass filtering (DC-absolute respiratory measurement).

### 2.4. Data Processing

Since the signals were acquired in controlled conditions, with minimal movements and artifacts, and after confirmation by visual inspection, additional filtering of the ECG signals was not applied. For the respiration signal only, a Chebyshev low-pass filter of the 4th order was applied to smooth out the small jitters that occasionally appeared at the onset of expiration, but with no influence on the research results. We set the corresponding cut-off frequency to 1 Hz.

*RRI* and *BBI* were obtained by the automatic detection of R peaks in the ECG recording, while the onsets of inspirations were taken as *BBI* marks. We performed this procedure by means of the Pick Peak tool of the Origin software (Microcal, Northampton, MA, USA). Rare missed and false-detected R peaks were manually added and removed, respectively. RR intervals were then simply calculated by subtracting the time coordinates of successive R peaks, and BB intervals were calculated by subtracting the time coordinates of successive inspiration onsets.

All programs implementing methods described in this work are original and were developed in MATLAB 2010a (MathWorks Inc., Natick, MA, USA).

### 2.5. Statistics

Statistical tests were performed using either a Friedman ANOVA, a post hoc Wilcoxon matched-pairs test, or the Mann–Whitney U test (Statistica 8.0, Stat Soft Inc., Tulsa, OK, USA). Nonparametric tests were used because we had a mixture of quantities (linear and angular). The Wilcoxon matched-pairs test was used as a post hoc test because analogous (matched) measurements were performed on the same individuals in different experimental conditions for each of the four states.

### 2.6. Two Types of Correlations

In our previous work, we described our approach to calculating *PRQ,* including both its integer and non-integer parts [16]. Here, we shall present only the basics of the procedure.

Let *R_j_, R_j+1_*, …, *R_j+k_* be the occurrences of R peaks positioned within the *i*th BB interval, itself lasting between the occurrences of two successive respiratory minima, *B_i_* and *B_i+1_*. Then, the integer number of RR intervals belonging to the *i*th BB interval is obviously *PRQ_int_* = *k*. These “integer” RR intervals, according to previously introduced peak symbols, could be denoted as *R_j_R_j+1_, R_j+1_R_j+2_, …, R_j+k-1_R_j+k_*. However, if we want additional precision in calculating *PRQ*, two “border” non-integer parts of two RR intervals, partitioned with *B_i_* and *B_i+1_*_,_ should also be taken into account. If we mark them with *b1* and *b2*, they can be calculated as:$$b1(i)=\frac{R_{j}-B_{i}}{R_{j}-R_{j-1}}\;\mathrm{and}\;b2(i)=\frac{B_{i+1}-R_{j+k}}{R_{j+k+1}-R_{j+k}}.$$

Besides gaining additional precision, it should be pointed out that in this way, none of the RR intervals is omitted from the count and their total number is conserved since *b2(i)* and *b1(i + 1)* add up to 1:$$b1(i+1)+b2(i)=\frac{R_{j+k+1}-B_{i+1}}{R_{j+k+1}-R_{j+k}}+\frac{B_{i+1}-R_{j+k}}{R_{j+k+1}-R_{j+k}}=1.$$

The final formula for *PRQ* can now be written as:*PRQ* = *PRQ_int_* + *b1* + *b2*.(1)

Moreover, Equation (1) allows a simple and direct estimation of the mean *RRI* within the *i*th BB interval:(2)$$mRRI(i)=\frac{BBI(i)}{PRQ(i)}.$$

By using Equation (2), we were able to examine two types of correlations having *BBI* as their common factor: *BBI* vs. *PRQ* and *BBI* vs. *mRRI*. Although, according to Equation (2), these two correlations are not independent, their physiological interpretations are not identical but instead somewhat complementary; their graphic presentations offer different information that could be extracted:The degree of positive correlation between *BBI* and *PRQ* determines whether and to what extent the number of intra-*BBI* RR intervals increases with the increase in *BBI*;The degree of positive correlation between *BBI* and *mRRI* indicates whether and to what extent the mean intra-*BBI* RR interval is increasing with the increase in *BBI*.

Both of these correlations could easily be assessed by studying their corresponding scatter diagrams in each of the test subjects and in each of the four experimental conditions.

#### 2.6.1. Correlations between *BBI* and *PRQ*

Examples of the *BBI* vs. the *PRQ* scatter plots are presented in Figure 1. The presented examples point to two types of point distributions for the four physiological states that appear in these scatter plots. The first type is characterized by points tending to aggregate around the integer values of *PRQ* (Figure 1a); the second type is characterized by a more continuous distribution of points (Figure 1b). Moreover, these examples indicate, and the systematic calculations confirm, that these correlations are relatively high—Pearson’s coefficient spanned the range of 0.4682 < *r_p_* < 0.9977; in all, 80 correlations were seen (20 subjects × 4 states). In addition, they were all highly significant: 4.8448 × 10^−302^ < *p* < 9.1062 × 10^−8^.

There are two CRC parameters that can be linked to the geometric properties of (*BBI*, *PRQ*) clouds of points. If point *PRQ^st1^* > *PRQ^st2^*, i.e., the heart rate per *BBI* in the state *st1* (of a subject) is found above the same quantity in *st2* for the respective length of *BBI*, it is possible to say that an increased heart rate per *BBI* points to greater oxygen demand in that state (static property). However, if the linear regression slope of that cloud is greater than that of the other, where $\frac{\Delta PRQ^{st1}}{\Delta BBI}>\frac{\Delta PRQ^{st2}}{\Delta BBI}$, this would mean that a greater increase in the number of *RRI* per *BBI* is present for the same increase in *BBI* (a dynamic property). Usually, the directions of both static and dynamic parameters appear simultaneously (clouds with greater slopes appear above the ones with smaller slopes; see Figure 1); however, in a small number of cases, variations also occurred (not shown).

The main goal of this paper is to explore whether only these two types of scatter plots, whether quantized or continual, that reflect two CRC operational modes (regimes) prevail among the eighty cases, or whether there are more than two types, or even if there is a more or less continual series of scatter-plot types. A better separation of the two types will be possible when the correlations between the *BBI* and *mRRI* are studied and presented later. Our secondary goal is to explore whether and in what way these operational modes are related to the degree of P-synchronization (so named because an analysis of *PRQ* was used), determined by calculating the average value and standard deviation of the *b1* parameter. 

#### 2.6.2. Correlations between *BBI* and *mRRI*

In this system of reference equations, *PRQ* = const is represented by a straight line passing through the reference origin (0, 0); therefore, we were in a position to study how the scattered (*BBI* and *mRRI*) points positioned themselves in relation to these concurrent lines in more detail.

When all 80 scatter plots (20 subjects × 4 states) of *BBI* vs. *mRRI* were examined visually, two types could also be discerned—one seemingly at random, with, apparently, a very low (*BBI*, *mRRI*) correlation, while the other type consisted of several clusters, each displaying a high intra-cluster correlation of points, although here, because of this co-existence of clusters, the overall correlation was also low. Obviously, in these cases, the overall correlation has little physiological significance, while the correlation within each of the clusters presumably does have significance. Typical examples of these two scattering subtypes are presented in Figure 2a (one subject is in a supine position with spontaneous breathing (Subject #1, supin)) and Figure 2b (a different subject, standing, using 0.1 Hz slow breathing, Subject #2, (stand01)). The quasi-linear clusters of the Figure 2a subtype consisted of points lying in the vicinity of the concurrent straight lines, as for *PRQ_int_*.

The importance of calculating both the integer and non-integer parts of *PRQ* is displayed in Figure 3b,c, wherein the *PRQ* values were rounded to the nearest integer, according to the following formula:
(3)$${\left(mRRI\right)}^{r}(i)=\frac{BBI(i)}{round(PRQ(i))},\;i=1,\;2,\;\ldots \;$$

As can be seen, by only using integers only, all the scattering information is thereby lost, rendering the two types (*BBI* and *mRRI*) of scatter plots indiscernible.

In order to explore the properties of different scatter-plot subtypes, we need to establish an objective assessment of their number by finding quantitative measures for ascertaining the degree of aggregation of the points around the concurrent *PRQ_int_* lines. After calculating these measures for all 80 existing scatter plots, the distribution of parameter values, such as histograms or probability density estimates (PDE, available in MATLAB) can be used in order to determine the number and boundaries of different scatter-plot subtypes.

##### Aggregation Measures

(a) In order to devise a quantitative measure for the intensity of aggregation, the first idea would be to perform certain algorithmic clustering of all points, regardless of the scattering pattern. However, in these plots, because of the integer nature of *PRQ_int_*, it is reasonable to assume that boundaries between clusters are given in advance and are fixed; for each concurrent line, *PRQ_int_* = const = *n*, *n* = 1, 2, …, the corresponding cluster should be spreading across a sector defined by [*n* − 0.5, *n* + 0.5], as indicated by the dashed lines in Figure 3a. Under these circumstances, automatic clustering, depending on the type of applied algorithm, could, in some cases, define different cluster boundaries, not to speak of the irregular clustering that could potentially appear in plots where aggregation is low and where scattering acquires a random appearance. Bearing all this in mind, and having defined cluster borders, as in Figure 3a, we can apply them to any scattering type and calculate the coefficient of linear correlation (*r^ic^*) within the (*ic*)*th* cluster. All *r^ic^* data within each scatter plot can be averaged for all *nc* clusters, thus assigning one quantitative aggregating property per plot, i.e., per subject in a particular state:$$r^{av}=\frac{1}{nc}\sum_{ic=1}^{nc}r^{ic}.$$

However, in the case of random scatterings, these imposed cluster borders, although mathematically correct, may potentially cause a biased elevation of *r^ic^*, since the borderlines would artificially cut through the random scattered field. Although the quantitative effect of this bias has yet to be assessed using real data, this measurement technique proved not to be able to discern the scatter plots, neither between the various states nor between the different degrees of aggregation (Table 1).

(b) For this reason, we tried a parallel approach, based on the following idea. First, we consider a series of gradually widening sectors within each cluster, defined by [*PRQ_int_* − *w*, *PRQ_int_* + *w*], where *w* stands for half of the current sector width [0 < *w* < 0.5]. Then, if *N_tot_* denotes the total number of points lying within a cluster defined with [*PRQ_int_* − 0.5, *PRQ_int_* + 0.5], by counting the percentage of points, *P_w_* = *N_w_*/*N_tot_* × 100, that lie within these gradually widening sector borders and by plotting *P_w_*(*w*), we obtain a monotonously ascending curve, acquiring values between 0 and 100, since *P_w_*(0) = 0 and *P_w_*(0.5) = 100. The intensity of aggregation around *PRQ_int_* is then reflected in the pace of the rise in *P_w_*(*w*), and the area under this curve could be used as a targeted measure. However, as it has no extreme points (except for statistically insignificant ripples), and, more importantly, since the change in this area (or, alternatively, the *P_w_*(*w*) mean value) is not as sensitive to the intensity of aggregation, it is more convenient to observe the difference *P_d_*(*w*) = *P_w_*(*w*) − *P_r_*(*w*), where *P_r_*(*w*) stands for the analogous function, which is valid in the case of uniformly distributed points. Obviously, *P_r_*(*w*) is a straight line with the same border values as *P_w_*(*w*). The more pronounced the aggregation of the points, the more *P_w_*(*w*) and *P_r_*(*w*) differ, while, in the case of a quasi-random distribution, these two dependencies become similar. In most cases, a significant (non-ripple) maximum appears in *P_d_*(*w*), and its position could also be used as a measure for the degree of aggregation (although it shows an inverse one). However, a more statistically reliable quantity is simply an average of *P_d_*(*w*) over the range *w* = [0, 0.5]:(4)$$P_{w}^{av}=\frac{1}{n_{w}}\sum_{i=1}^{n_{w}}P_{w}(w_{i});P_{d}^{av}=\frac{1}{n_{w}}\sum_{i=1}^{n_{w}}P_{d}(w_{i}),$$
where *n_w_* stands for the number of discrete widening sector widths. In a small number of cases (scatter plots), negative values of *P_d_^av^* appeared, corresponding to the aggregation of points around *PRQ_int_* ± 0.5.

In order to avoid ripples and obtain curves that are as smooth as possible, we shall apply an approach based on the simultaneous widening of sectors for all clusters in a scatter plot. If *N_w_^ic^*(*w*) and *N_r_^ic^*(*w*) stand for an experimentally determined and randomly positioned number of points, respectively, as found within the sector of width, *w*, for the cluster *ic*, and *N_tot_^ic^* corresponds to the total number of points in that cluster, then, by simultaneously widening the sectors for all clusters, we obtain an equation for the whole scatter plot:(5)$$P_{w}^{tot}(w)=\frac{\sum_{ic=1}^{nc}\left(N_{w}^{ic}(w)\right)}{\sum_{ic=1}^{nc}N_{tot}^{ic}}\times 100$$
and
(6)$$P_{d}^{tot}(w)=\frac{\sum_{ic=1}^{nc}\left(N_{w}^{ic}(w)-N_{r}^{ic}(w)\right)}{\sum_{ic=1}^{nc}N_{tot}^{ic}}\times 100,$$
where *nc* stands for the number of clusters.

Finally, the averaging of all *P_w_^tot^*(*w*) and *P_d_^tot^*(*w*) values across *w* can be performed by applying Equation (4) to Equations (5) and (6):(7)$$P_{w}^{avtot}=\frac{1}{n_{w}}\sum_{i=1}^{n_{w}}P_{w}^{tot}(w_{i})$$
(8)$$P_{d}^{avtot}=\frac{1}{n_{w}}\sum_{i=1}^{n_{w}}P_{d}^{tot}(w_{i}).$$

As an example, two cases of *P_w_^tot^*(*w*) and *P_d_^tot^*(*w*), one for a scatter plot with an aggregation of points, and the other for a random distribution, are presented in Figure 4a,b, respectively.

(c) Since it is desirable to have and test as many measures as possible, we propose an additional measure for the intensity of aggregation. Namely, for a particular scatter plot, the sector half-width *w_max_*, where the maximum of *P_d_^tot^* is positioned, could serve as an inverse measure of the intensity of aggregation (Figure 4a).

(d) As the fourth measure for the intensity of aggregation, these also being inverse, one can compute for each cluster *ic*, containing the *n_ic_* data, determined with the boundaries *PRQ_int_* ± 0.5, conventional standard deviation, using the MATLAB command “std”, around *PRQ_int_* of the corresponding intra-cluster sets of data: (*PRQ*(*i*))*^ic^* = (*BBI*(*i*))*^ic^*/(*mRRI*(*i*))*^ic^*, *i* = 1, …, *n_ic_*, is denoted as (std(*PRQ*))*^ic^*. We then average them across the clusters, with:$${\left(std(PRQ)\right)}_{av}=\frac{1}{n_{c}}\sum_{ic=1}^{nc}{\left(std(PRQ)\right)}^{ic}.$$

(e) The fifth measure can be derived by the mathematical modeling of *P_w_^tot^*(*w*) using a one- or two-parameter model. After the nonlinear fitting of the model equation on all experimentally obtained curves, the distribution of parameter values can be employed to determine the number and boundaries of the different scatter-plot subtypes.

An initial idea was to model the measures with a one-parameter exponential model:$$P_{wm}^{tot}(w)=\pm 100\times \left(1-e^{-\alpha w}\right),$$
where plus and minus signs were applied if *P_w_^tot^* was lying above or below *P_r_*, respectively (although mixed cases were also detected). However, since, in the case of *P_w_^tot^* ≈ *P_r_* the model did not fit well with the data, we used its two-parameter generalization by substituting the constant first term with an exponential one:(9)$$P_{wm}^{tot}(w)=100\times \left(e^{-\alpha_{1}w}-e^{-\alpha_{2}w}\right).$$

Two examples of these fittings are presented in Figure 5.

### 2.7. Cardio-Respiratory Synchronization

#### 2.7.1. *β*1. Locking

Our method offers an original approach to studying CR synchronization. Namely, instead of utilizing a number of standard methods, such as those based on the Hilbert transform [4,19,20,21,22], we can use the statistical properties of the non-integer parts of *PRQ*, denoted previously as *b1* and *b2*. Specifically, we let *b1*(*i*), *i* = 1, …, *N*, be the ensemble of the *b1* values calculated from *N* BB intervals during a recording session. Although this measure is, strictly speaking, not an angle, its statistical properties can be used as a relatively good substitute for angles, since it possesses a periodic nature wherein the range [0, 1] stands instead of [0, 2π] or [−π, π], the latter being more suitable for MATLAB commands. This periodicity is based on the fact that both *b1* = 0 and *b1* = 1 signify the same thing—namely, that an R impulse is synchronized with the onset of the respiratory cycle. However, in order to apply angular statistics [23], it is necessary to map *b1* onto an angle (e.g., denoted as *β1*), *β1*(*i*) = 2 π *b1*(*i*) − π, calculate the angular mean value, and map the average back to the [0, 1] range:(10)$$\beta 1a=\arctan\frac{\sum_{i=1}^{n}\sin(\beta 1(i))}{\sum_{i=1}^{n}\cos(\beta 1(i))};b1a=\frac{\beta 1a+\pi }{2\pi }.$$

On the other hand, by calculating histograms approximating the distribution of the number of *BBIs* over *b1*(*i*), or the corresponding probability distribution estimate, PDE(*b1*), we can extract two kinds of information:*b1_m__ax_*, the value of *b1* where the maximum of PDE or its histogram occurs. Using this analogy, *b1_max_* signifies the most probable “phase shift”, or, more precisely, the most probable part of the RR interval between respiration onset and the first occurring R impulse. However, since, in some cases, there is more than one PDE or histogram maximum, it is more reliable to use the angular mean value, according to Equation (10);histogram-derived standard deviation may be denoted with *b1_hst_* and may be used as a reciprocal measure for the degree of “*b1* locking” or “*β1* locking” (further details will be explained in Section 2.7.2). Note that it is not necessary to compute analogous quantities for *b2*, since, except for their first and last appearances in the signal, their relationship is complementary: *b1*(*i*) + *b2*(*i* − 1) = 1, *i* being the *i*th BB interval.

#### 2.7.2. Circular Correction

Circular correction is a procedure used to calculate the correct mean value and correct standard deviation of an angular (periodic) random variable, because classical (linear) methods, in some cases, yield incorrect results. It is particularly efficient in the case of uni-modal probability distributions that have shifted away from the central position. Let us imagine a wide quasi-Gaussian distribution, in the range of [−π, π], which is being gradually dislocated to the left. If shifted enough, because of the distribution’s periodic nature, its left side would gradually appear on the right end of the range. Consequently, if calculated in a classical manner, these parts of the distribution that are appearing would cause the mean value to be shifted to the right of the peak, while the standard deviation would be increased. If the shift mounts to π (“contra-phase”), we obtain a U-shaped profile, where the classical mean value becomes approximately zero, (instead of being near ±π), while the standard deviation becomes maximally (erroneously) increased. In order to correct this discrepancy, a series of calculations with gradually shifted distribution should be performed, and the resulting series of standard deviations (*SD*) should be recorded. Then, the minimal value of all obtained *SD* values (*SD_min_*) should be taken as correct, while the amount of the shift corresponding to *SD_min_* should be considered to be the true mean value. The position corresponding to *SD_min_* may be considered to be “centered”. Circularly corrected averages do not differ greatly from the angular mean values, while standard deviations do differ, and are more accurate than angular ones, since the latter are accurate only for small angles.

Let us consider the case of parameter *b1* and use histogram terms. If a histogram were constructed from all the *b1* values collected during a recording session, and if Δ*b1*(*j*) and *N*(*j*) are the *j*th histogram bin, where *j* = 1, …, *n*, and its corresponding value, respectively, then the gradual shifts could be termed as Δ*b1*((*j* + *k* − 1) mod *n* + 1) and *N*((*j* + *k* − 1) mod *n* + 1), *k* = 0, …, *n* − 1, where *n* stands for the total number of bins and mod for the modulus operation (in other words, a series of *n* circular permutations of bins are being generated). Then, the histogram-wise average value of *b1* (*b1_ha_*(*k*)) and its standard deviation (*b1_hst_*(*k*)) for shift *k* could be calculated as:$$b1_{ha}(k)=\frac{\sum_{j=1}^{n}N\left((j+k-1)\mathrm{mod}n+1\right)b1^{\prime }\left((j+k-1)\mathrm{mod}n+1\right)}{\sum_{j=1}^{n}N\left((j+k-1)\mathrm{mod}n+1\right)}$$
and
$$b1_{hst}(k)=\sqrt{\frac{\sum_{j=1}^{n}N\left((j+k-1)\mathrm{mod}n+1\right){\left(b1^{\prime }((j+k-1)\mathrm{mod}n+1)-b1_{ha}(k)\right)}^{2}}{\sum_{j=1}^{n}N\left((j+k-1)\mathrm{mod}n+1\right)}},$$
where *b1′*((*j* + *k* − 1) mod *n* + 1) stands for the *b1* value at the ((*j* + *k* − 1) mod *n* + 1)th bin.

The circularly corrected value of the *b1* standard deviation can then be taken as:

(*b1_hst_*)*_c_* = min(*b1_hst_*(*k*)), *k* = 0, …, *n* − 1.

## 3. Results

### 3.1. Correlations between BBI and PRQ

The correlation between *BBI* and *PRQ*, obtained for all twenty subjects in four experimental conditions, is presented in Figure 6.

#### 3.1.1. Positions of Points and Linear Regression Slopes

According to this group result, in all four states, there is a generally positive correlation between *BBI* and *PRQ*, meaning that the number of intra-*BBI* RR intervals increases with the increase in *BBI*. Moreover, slow breathing seems to have little or no effect on the slopes of these correlations, but body posture does. In other words, the slopes indicate that slow-paced 0.1 Hz breathing has little or no effect on the increase rate of the number of *RRI*s per *BBI*, with respect to the increase rate of the *BBIs* in spontaneous breathing (see Figure 6, where black and green clouds are positioned as a continuation of the blue and red clouds, respectively), while the vertical positions of the clouds mean that, for the same *BBI*, in stand and stand01, there is a greater number of *RRI*s per *BBI* than in the supin and supin01 states (Figure 6, where red and green clouds are positioned above the blue and black ones). In addition, the increased slopes in these two states are characterized by an increased sympathetic drive, meaning that, statistically speaking, the number of *RRI*s for each *BBI* rises more quickly with an increase in *BBI* in stand and stand01 than in supin and supin01.

#### 3.1.2. Statistics of Slopes and Pearson’s Coefficient

The slopes of the *BBI* vs. *PRQ* group regression lines differed significantly for each tested pair of states (Figure 10a, Table 1). However, it is evident from the bar diagram in Figure 10a that these slopes are greater for the stand and stand01 body postures than for the supin and supin01 body postures.

Although it is not as obvious when looking at the bar heights in Figure 10b, the group test results show that Pearson’s coefficient of the linear correlation between *BBI* and *PRQ*, due to the low group standard deviation levels, differed significantly between the two breathing regimes, while they were not different within each of the regimes (Figure 10b, Table 1). In other words, slow breathing significantly decreased this correlation, regardless of the body posture, although the correlation coefficient remained relatively high.

### 3.2. Correlations between BBI and mRRI

After inspecting all eighty scatter plots, five main subject subgroups could be distinguished:Subjects where the scatter plots had visible clusters of high within-cluster correlations of *BBI* vs. *mRRI* data in all four experimental conditions (Figure 7a).Subjects showing scatter plots displaying high within-cluster correlations only when in a supine position, regardless of the breathing regime (Figure 7b).Subjects with scatter plots displaying high within-cluster correlations, but only in the regime of spontaneous breathing, regardless of the body posture (Figure 7c).Subjects showing scatter plots without any clustering of the data, regardless of the body posture or breathing regime (Figure 7d).Subjects that had scatter plots displaying high within-cluster correlations, seen only in one of the four experimental conditions or other combinations of clustering cases (not shown).

The more complex nature of *BBI* vs. *mRRI* scatter plots compared to *BBI* vs. *PRQ* scatter plots where multiple clusters appear, which show high intra-cluster and low integral correlation, renders inadequate integral correlation analysis, such as that seen in the approach applied to the *BBI* vs. *PRQ* scatter plots.

Additional scattering-group properties can be obtained if the scatter plots of all twenty subjects in all four states are superimposed onto one plot (Figure 8).

It can be seen from Figure 8 that those cases with high intra-cluster correlations, where the points tend to aggregate around the *PRQ_int_* lines, appear predominantly in the region of longer *mRRI* values (roughly > 1 s), which signifies that the zone is undergoing vagal domination.

#### 3.2.1. Dynamic Behavior of (*BBI* and *mRRI*) Points

When looking at Figure 2, Figure 3, Figure 7 and Figure 8, one can hardly avoid querying the dynamics of one particular observed (*BBI* and *mRRI*) point in the presented scatter plots. Some of the cases are shown in Figure 9, where plots recorded from a subject with pronounced clustering (Figure 9a,b), as well as another with random scatter-plot behavior (Figure 9c,d), are depicted.

It is clear from Figure 9a that in the case of pronounced clustering, the *PRQ* values exhibit a more stable behavior in the time plot, with one *PRQ_int_* value serving as a “base” (in this case, *PRQ_int_* = 5), wherein the system tends to remain for a few BB intervals and from which the bounces to neighboring clusters are taken (Figure 9a,b). These crossings end, in most cases, on the next cluster (higher or lower alike, *PRQ_int_* ± 1), while the jumps to *PRQ_int_* ± 2 are less frequent, although they are still present (two of them can be seen in Figure 9a as the transitions from *PRQ_int_* = 5 to 3 and back, shown as black lines). While remaining on the “base” cluster, the (*BBI* and *mRRI*) point exhibits random, erratic jumps between the black circles, while still staying within *PRQ_int_* = 5 (Figure 9b, red intra-cluster jumps). In addition, after crossing to the neighboring cluster, the system rarely stays in it for longer than one BB interval. In contrast to that finding, scatter plots with random behavior show no base *PRQ_int_* value, and the randomness of the system trajectory extends to the whole plot (Figure 9c,d).

#### 3.2.2. Statistics of Aggregation Measures

When observed visually, all 80 scatter plots could be identified as belonging to either those plots with an aggregation of points around the *PRQ_int_* lines (AG), or those with a random distribution of points (RD), similar to the cases presented in Figure 2a,b. In total, there were 28 AG scatter plots (15 supine; 4 standing; 8 supine with slow 0.1 Hz breathing; one standing with 0.1 Hz breathing) and 52 RD scatter-plots (5, 16, 12, and 19 in the four states, respectively). This visually performed classification was used to form the AG-RD bar diagrams (see Figure 10d,f,h,j,l,n on the right side). Only the cluster-averaged Pearson’s coefficient (*r^av^*; Figure 10c,d; Table 1) was neither able to discern the four states, nor to differentiate the AG from the RD scatter-plots (although in the latter case, the Mann–Whitney U test showed a tendency toward significance; *p* = *0.085*). All other five measures successfully passed the Mann–Whitney U test in terms of separating AG from the RD scatter plots, which was the main goal of this paper. However, when their ability to separate the four experimental states was tested, the measures showed mixed results, with the number of successes ranging from 3 to 5 out of 6 pairs (see Figure 10c,e,g,i,k,m, on the left side; Table 1).

**Figure 10 bioengineering-10-00180-f010:**
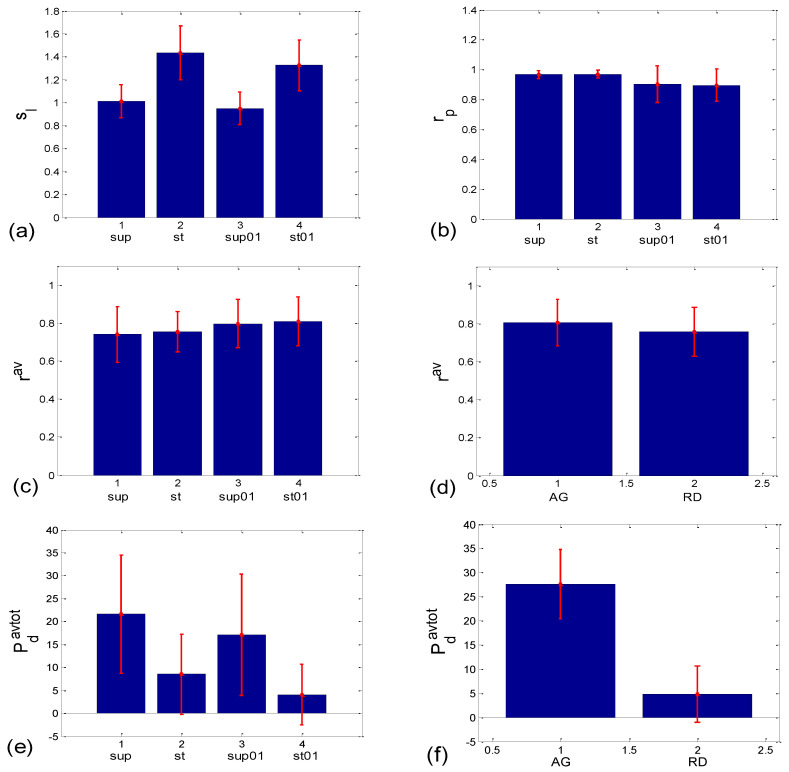
Bar plots showing the group mean ± standard deviation of different correlation or aggregation measures. (**a**,**b**) Related to *BBI* vs. *PRQ* correlations: (**a**) slopes of linear regression lines (*s_l_*); (**b**) Pearson’s coefficients (*r_p_*); (**c**–**n**) quantities related to *BBI* vs. *mRRI* correlations which measure the degree of aggregation of points around *PRQ_int_* lines. Left side—with respect to four states; right side—with respect to the scatter plots characterized with aggregation (AG) or a random (RD) distribution of points. The statistical test results are displayed in Table 1.

Incidence histograms of the proposed six quantitative aggregation measures are presented on Figure 11. 

### 3.3. Cardio-Respiratory Synchronization

#### 3.3.1. *b*1 Averaging

Let (*b1a*)*_s_* be the *b1* equivalent of angular average (*β1a*)_s_ of all *β1*(*i*) values, mapped from the recorded *b1*(*i*) values obtained during a recording session from subject *s*, according to Equation (10). Let the further (*b1a*)*_g_* be the group average, inverse-mapped from the group angular average (*β1a*)*_g_* of all (*β1a*)*_s_* values across the tested subjects:(11)$${\left(\beta 1a\right)}_{g}=\arctan\frac{\sum_{s=1}^{N_{s}}\sin\left({\left(\beta 1a\right)}_{s}\right)}{\sum_{s=1}^{N_{s}}\cos\left({\left(\beta 1a\right)}_{s}\right)};{\left(b1a\right)}_{g}=\frac{{\left(\beta 1a\right)}_{g}+\pi }{2\pi },$$
where *N_s_* stands for the number of subjects. This “double angular” average of all the (*b1*(*i*))*_s_* recorded values proved to be more sensitive to differences, either between states or between the AG-RD scatter plot subtypes, than the linear averaging of both *b1max* and *b1*(*i*)*_s_* (Figure 12, where a comparison with *b1max* is shown as an example). This result also corroborates the assumed angular nature of *b1*. However, this particular property of *b1* caused some discrepancy in terms of the obvious and sizeable differences between the group mean values (bar heights) in Figure 12c and the corresponding test results presented in Table 1. Namely, although the Friedman ANOVA showed a significant value (*p* = 0.001), the Wilcoxon matched-pairs tests resulted in 5/6 pairs having insignificant (*b1a*)*_g_* differences. This is probably due to the fact that this test does not take into account the angular nature of *b1*; rather, the quantities contained in the 20 × 4 matrix are being treated as linear random variables, while the bar heights in Figure 12c were obtained using angular statistics, which is, in our view, the only approach that is adequate for angular quantities.

Numerical results for the linear mean ± standard deviation of *b1max* in the four states (Figure 12a) were as follows: 0.349 ± 0.087 for supine; 0.381 ± 0.282 for standing; 0.384 ± 0.109 for supine with 0.1 Hz breathing; 0.336 ± 0.183 for standing with 0.1 Hz breathing. The Friedman ANOVA yielded insignificant differences between the tested values (*p* = 0.085). When the *b1max* for the scatter plots with aggregation (AG) and the random distribution of points (RD) was compared (Figure 12b), resulting in 0.353 ± 0.116 and 0.367 ± 0.207, respectively, the Mann–Whitney U test did not find a significant difference between them (*p* = 0.333). On the other hand, according to Figure 12c, group mean ± standard deviation, for (*b1a*)*_g_* values in supin and supin01 are higher than in stand and stand01 (0.347 ± 0.146 and 0.393 ± 0.169, vs. 0.154 ± 0.215 and 0.206 ± 0.231, respectively). We have to bear in mind that standard deviations are overestimated when a formula for angular versions, similar to Equation (12), is applied since it is only accurate for small angles. Therefore, it is advisable to use circularly corrected standard deviations instead, as in Figure 12c,d.

The results presented in Figure 12c show that in states with vagal domination, less than approximately 2/3 of an average RR interval length, shared between two successive BB intervals, is located before the respiration onset, while slightly more than 1/3 of the interval length occurs after the onset. When the sympatho-vagal balance shifts to sympathetic domination (as in stand and stand01), the average RR interval is moved to a greater “R pulse—breathing onset” synchrony, resulting in only approx. 15–20% of the RR interval length occurring after the beginning of inspiration, with 80–85% occurring before it. In addition, the result presented in Figure 12d shows that in the AG scatter plots, the double angularly averaged (*b1a*)*_g_* values point to a greater part of the shared RR interval occurring after the respiration onset in AG than in the RD scatter plots (0.354 ± 0.113 vs. 0.239 ± 0.232, respectively).

##### 3.3.2. *b*1 Locking and Circular Correction

As stated earlier, the standard deviation of (*b1*(*i*))*_s_*, collected from all the BB intervals *b1*(*i*), *i* = 1, …, *N* recorded during a session with the subject, *s*, may serve as an inverse measure of the *b1* locking. This is analogous with but not identical to conventional phase locking. The difference between the two notions lies in the fact that, although periodic, *b1* changes its “frequency” between two successive values, in correlation with changes in the instantaneous values of the RR intervals. However, while angular averaging is more accurate than linear averaging, angular standard deviation, according to Equation (12):(12)$$\alpha_{st}=\sqrt{\arctan\frac{\sum_{i=1}^{n}\sin^{2}\left(\alpha_{a}-\alpha (i)\right)}{\sum_{i=1}^{n}\cos^{2}\left(\alpha_{a}-\alpha (i)\right)}},$$
where *α_a_* stands for the angular average, tends to overestimate the real dispersion of angles since it is a direct generalization of the linear formula and is accurate only for small deviations from the mean angle. Therefore, we shall use circularly corrected histogram-derived standard deviation (*b1_hst_*)*_c_*, as defined in Section 2.7.2 (as an example, the result for one subject is shown in Figure 13).

As can be seen in Figure 14a, the circularly corrected (*b1_hst_*)*_c_* value in supine postures and in supine postures with slow 0.1 Hz breathing (0.146 ± 0.062; 0.169 ± 0.075, respectively) is lower than in standing postures with the same breathing regimes (0.215 ± 0.049; 0.231 ± 0.029, respectively), signifying that a greater degree of *b1* locking is occurring in states with vagal domination. When compared using the Wilcoxon matched-pairs test, a significant difference in the degrees of *b1* locking was observed in the four pairs of states, while in stand vs. stand01 and supin01 vs. stand01, they were not different (Table 1, although in stand vs. supin01, there was a tendency toward significance, *p* = 0.062). An analogous phenomenon was recorded in scatter plots with aggregation and a random distribution of points; they were lower in AG (0.113 ± 0.036) and higher in RD (0.232 ± 0.031, Figure 14b), showing a greater degree of *b1* locking in the AG scatter plots. The Mann–Whitney U test proved that this difference was significant (*p* = 0.000, Table 1).

**Table 1 bioengineering-10-00180-t001:** Summarized results of the statistical tests.

Corr Type	Measure	Friedman ANOVA	Post hoc W Sup-St	Post hoc W Sup- Sup01	Post hoc W Sup- St01	Post hoc W St- Sup01	Post hoc W St-St01	Post hoc W Sup01- St01	MWU AG vs. RD
*BBI* vs. *PRQ*	*s_l_*	**0.000**	**0.000**	**0.021**	**0.000**	**0.000**	**0.002**	**0.000**	
	*r_p_*	**0.000**	0.351	**0.023**	**0.001**	**0.015**	**0.001**	0.970	
*BBI* vs. *mRRI*	*r^av^*	0.323							*0.085*
	*P_d_^avtot^*	**0.000**	**0.002**	*0.067*	**0.000**	**0.028**	**0.037**	**0.000**	**0.000**
	*w_max_*	**0.005**	**0.027**	0.644	**0.001**	0.171	0.380	**0.011**	**0.000**
	(std(*PRQ*))_av_	**0.000**	**0.000**	**0.010**	**0.000**	**0.021**	0.332	**0.004**	**0.000**
	α_1_	**0.000**	**0.010**	**0.044**	**0.000**	*0.086*	0.145	**0.000**	**0.000**
	α_2_	**0.000**	**0.001**	0.117	**0.000**	**0.044**	*0.052*	**0.000**	**0.000**
*b1* synch.	(*b1a*)*_g_ **	**0.001**	0.204	0.191	*0.052*	0.145	0.279	**0.019**	*0.069*
	(*b1_hst_*)*_c_*	**0.004**	**0.001**	**0.048**	**0.000**	*0.062*	0.117	**0.010**	**0.000**

* Results were obtained with conventional nonparametric tests, offered by the available software, which does not treat data in the ensembles as angular random variables. However, since the angular mean values of states show greater inter-state differences than conventional linear ones (Figure 12a,c), special angular tests would probably result in significant differences. Values in bold type show a significant *p* < 0.05, while values in italics show a tendency of 0.05 < *p* < 0.1.

## 4. Discussion

### 4.1. Correlation between BBI and PRQ

In terms of the positive *BBI* vs. *PRQ* correlations (Figure 6), this result seems logical at first glance: the longer the *BBI*, the greater the number of *RRI*s within it. However, this could be an oversimplified view due to the fact that, for the same value of *BBI*, there is a higher number of *RRI*s within the *BBI* in stand and stand01 (active standing-specific sympathetic dominance), than in supin and supin01 (the green and red dots are above the blue and black ones (see Figure 6)). Therefore, after a transition from the supine position, there is a greater number of heartbeats in the same *BBI* than at rest (in “stand”, the increased heart rate primarily regulates the central arterial blood pressure values by increasing the cardiac output, which ensures adequate oxygenation of the heart muscle and the CNS). This heart-rate response can be considered to be a consequence of the effect of gravity on blood distribution to the body’s periphery, while the combined increase in heart and breathing frequencies can be considered to be a consequence of increased metabolic activity in antigravity muscles and respiratory pump muscles (the response to relative hypoxia and relative hypercapnia during active standing, compared to when supine). We consider the greater degree of heart-rate acceleration compared to the degree of breathing-rate increase during active standing to be a consequence of the combined dominant effect of baroreflex and chemoreflex (as well as metaboreflex), which act synergistically on the heart rate, while the increase in *BBI* is a consequence of the activation of chemoreflex (as well as metaboreflex) [24]. This is a “static” analysis, and there is also a “dynamic” one: it is reflected in the fact that the slopes of red and green regressions (stand and stand01) are higher than those of blue and black (supin and supin01), which means that the number of *RRI*s per *BBI* increases faster with increasing *BBI* in stand and stand01 than in supin and supin01. This supports our hypothesis that multiple co-directional, synergistic mechanisms accelerate the heart rate during active standing, compared to supination.

Another particularly interesting result of our experiments is that the breathing regime, i.e., slow-paced 0.1 Hz breathing compared to spontaneous breathing, has no significant effect on the relationship between *PRQ* (i.e., the number of *RRIs* per *BBI*) and *BBI* (see Results, Section 3.1.1, or Figure 6), nor on the relationship between the dynamics of heart rate deceleration and *BBI* increase. With this result, we complete the hypothesis that the modulation of the two mentioned linear parameters of the cardiorespiratory coupling, “static” and “dynamic”, is dominantly regulated by heart-rate regulation mechanisms that are specific to active standing. These mechanisms ensure the modulation of the sympatho-vagal relationship, in favor of sympathetic dominance (the black points are an extension of the blue points, with the green the extension of the red ones (Figure 6)). It is interesting that even in the state considered for stand01, with the combined sympathetic and parasympathetic action on the sinoatrial node [17], the sympathetic action dominates. This confirms that orthostasis is the main stressor of heart-rate regulation mechanisms and is the dominant factor of CR coupling to be expressed by the proposed linear parameters.

### 4.2. Correlations between BBI and mRRI

Here, positive correlations were obtained as well (albeit the more complex, “striped”, or multi-clustered results only seen in the AG scatter plots), which means that, within one “strip”, i.e., cluster, with an increase in *BBI*, the average *RRI* within it also increases. This result indicates that in conditions where the AG type of *BBI* vs. *mRRI* relationship dominates, there is a functionally active mechanism that induces the same direction change—an increase in both *BBI* and *mRRI*. Hypothetically, this mechanism could also be sympathetic in nature, with its effect on the *BBI* regulation mechanism being indirect.

By applying several proposed measures for an estimation of the degree of aggregation of points around the *PRQ_int_* lines, we obtained roughly bimodal histograms for the scatter plots (Figure 11), which indicates the existence of only two of their types (AG and RD), and which further points to the existence of two modes of CR coupling organization, with relatively few transitional forms (because the measurements are not ideal, and nor are the modes, i.e., the peaks of the histogram are roughly separated). In addition, visual assessment (AG incidence: 15 in supin; 8 in supin01 vs. 4 in stand; 1 in stand01) was corroborated by the results in Figure 10e–n, Figure 12c,d, and Figure 14, where the bar heights in supin and supin01 were correlated with the ones in AG, while those in stand and stand01, correlated with the heights in RD. This strongly indicates that AG scatter plots prevail in states with vagal dominance. Our hypothesis, supported by the results from [16], is that with sympathetic activation, i.e., physical effort (active standing), the transition from an AG to an RD state implies a process of adaptation or a manifestation of adaptability ([25] and others (see, for example, [16])) to an ambient force (gravity), which is a muscularly and, consequently, metabolically more demanding physiological state.

It can be observed in Figure 8 that high intra-cluster correlations occurred, mostly when RR intervals longer than 1s were present (a slower heart rate), i.e., in the case of vagal predominance. Previous studies have also pointed to this *RRI* length as being specific for the vagal dominance of heart-rate control [24,26,27,28]. Therefore, the AG plots, which are mainly characterized by longer RR intervals (Figure 8) and which primarily appear in the supin and supin01 states, point to vagal dominance in terms of heart-rate regulation. *PRQ* quantization in conditions of vagal dominance is in accordance with the known trophotropic function of the vagus [29,30,31,32,33,34], i.e., the tendency to (re)store energy and its accumulation in states dominated by vagus control over the internal organs (i.e., sleep, [35,36,37,38,39]). The integer ratio of *BBI* and *RRI* could indicate the energetically ideal and “most economical” ratio of heart work (blood stroke volume) and respiratory pump work (ventilation), described as “ventilation-perfusion efficiency” [24,26], which is sufficient to meet the body’s energy needs for oxygen in conditions of constant energetic demand [40].

Regarding the relationship between the two CR regimes, identified with the AG and RD scatter plots on the one hand, and the strength of the CR coupling on the other hand, it has already been reported that the sympathetic activation induced by active standing reduces the strength of the causal dependence (directionality) from respiration to the RR interval series (Resp → *RRI*) [41]. By looking at the two types of scatter-plot distributions of (*BBI* and *mRRI*) points, where the AG type is more frequent in supin (15/20) and supin01 (8/20) than in the stand (4/20) and stand01 (1/20) states, and functions under the assumption that the AG type corresponds to a greater Resp → *RRI* dependence, then our results are in agreement with those reported in [41]. However, in order to corroborate this finding with a quantitative result, analyses that include a model, such as, e.g., closed-loop multivariate dynamic adjustment ((CLMDA), as applied in [41]), or some other causality-measuring procedure should be carried out.

### 4.3. CR Synchronization

In AG states (vagal dominance) approximately 2/3 of the average shared RR interval is within the previous *BBI* range and 1/3 of the length after the start of inspiration (Figure 12c,d; left bar). This locking is stronger than in the RD state (Figure 14a,b; left bar). In contrast, in RD plots with randomly scattered points, where the *PRQ* quantization is not so pronounced, the intercepted RR interval shifts to the left (relative to the inspiration onset), so that about 4/5 of the shared *RRI* occurs in the previous *BBI*, and only about 1/5 occurs after the start of inspiration (Figure 12c,d; right bar). This situation shows a weaker locking (a larger circularly corrected standard deviation (Figure 14a,b; right bar)). We, thus, take into account the fact that the beginning of *RRI* implies the beginning of the ventricular systole, followed by the same duration of ventricular diastole, followed by a new *RRI*, i.e., a new ventricular systole [24,26]. The measured “2/3 vs. 1/3” distribution of the truncated RR interval implies that most of the diastole (1/3 of *RRI*) is in the period of taking over the oxygen from the air (the inspiration part of *BBI*) during vagal dominance, which is typical for the AG type of the *RRI* vs. *PRQ* ratio. Since the oxygenation of the heart muscle is mostly in diastole [42], this means that the heart muscle has enough time for adequate oxygenation. The shift toward the “4/5 vs. 1/5” distribution in conditions when higher oxygen demands are expected, which is specific for RD conditions with sympathetic dominance (active standing), means that the heart in these conditions has less time for its own oxygenation, i.e., it functions in a state of relative oxygen deficiency with respect to the AG conditions of vagal dominance.

The results presented in Figure 7, Figure 8 and Figure 9 could also point to an explanation regarding the mechanism by which the CR system is able to produce integer *PRQ* peaks, as presented in our previous work [16]. It seems to be a dynamic process, maintained by continuous system regulation. When active, as in the case of high intra-cluster correlation (AG scatter plots), this regulation is apparently sufficiently strong that we can even speak of an “integer *PRQ* homeostasis”. Namely, from Figure 2, Figure 3, Figure 7, Figure 8 and Figure 9, it can be inferred that when the CRC regime is such that the (*BBI* and *mRRI*) points lie near a *PRQ_int_* line, if the *BBI* increases, it is followed by a corresponding *RRI* increase, and vice versa. It is reasonable to hypothesize that this adaptive phenomenon is a consequence of a joined integrative mechanism of regulation of a higher order, i.e., at a hypothalamic level [30]. Even if a (*BBI* and *mRRI*) point temporarily leaves a cluster, when returning back, it still follows the high-correlation rule (Figure 9a,b). This speaks in favor of the hypothesis that in the states characterized by high vagal dominance, there are different “step-like” regimes that could correspond to a set of defined, different, and discrete *BBI* vs. *mRRI* regimes corresponding to stable metabolic states, a sort of predefined CR software for stable metabolic needs with the lowest energy expenditure necessary for the cardio-respiratory supersystem.

Our approach, wherein *BBI* and *mRRI* emerge as temporally synchronized and concurrent parameters of CR coupling, offers a perspective by which to apply a wide palette of existing methods for calculating coupling strength and information transfer, especially because *mRRI* is a new and unexploited quantity. Since the presence of different scatter-plot subtypes points to two different CR regimes, in the future, it would be of interest to explore some of their specific properties, such as the existence and intensity of the mutual influences on *BBI* and *mRRI* (directionality), as well as the strength of cardio-respiratory coupling in AG and RD regimes. For this purpose, it is possible to use different strategies, such as Granger causality [43,44], the Kuramoto–Daido model coupling strength [45], conditional and transfer entropy [46,47,48,49], etc. The directionality of cardio-respiratory interaction (CRI) has been studied extensively in atrial fibrillation patients [44] and in infants [22], wherein the authors stated that the direction of coupling between the cardiovascular and respiratory systems varies with age within the first 6 months of life. The authors found an evolution from an almost symmetric bidirectionality to a nearly unidirectional interaction from respiration to the cardiovascular system. It is well known that bidirectional interactions between the respiratory system and the heart are present and can be quantified (e.g., [50]). However, regarding our approach, while it is easier to qualitatively assess the CR coupling strength by correlating it with this type of analyzed scatter plots (AG—stronger; RD—weaker), it is harder to infer the CR interaction direction, based on the characteristics of these two types of distributions. After decomposing *PRQ* into *PRQint*, *b1,* and *b2*, there are two candidates for such an analysis—*mRRI* and *b1*, which are both temporally concurrent with the *BBI* time series (*b2* is not usable because it is not independent of *b1*). However, due to the length of our present paper, we were in a position to postpone investigating the Granger directionality of these interactions for our future work. Still, even though the topic is interesting, the question of directionality does not respond to the question of fine anatomic–functional CR interactions, besides baroreflex (C → R) and respiratory sinus arrhythmia (R → C). It would be important to investigate the cardiovascular and respiratory systems from the integrative perspective, in their role as the physiologic state-dependent effectors of the higher-order neural networks that are responsible for CR integration. With respect to energy economics and other state-specific parameters, it is reasonable to hypothesize that the same integrative network alternatively (or jointly) potentiates the dominance (or equity in directional influence) of different effectors (C or R) for the correction of arterial blood oxygenation, pH, and the arterial blood level of CO_2_, with respect to the specific physiological state.

Cardiovascular and respiratory systems were traditionally regarded as weakly coupled oscillators [49]. However, Schafer et al. [2] developed a technique for detecting CR synchronization using the Hilbert transform to calculate the instantaneous respiratory phase and follow it in successive R peaks. They showed the existence of relatively long periods of up to 20 min of synchronized CR activity in young, healthy athletes. They found that phase synchronization is stronger in those subjects with weaker RSA and also concluded that the “phase locking of respiratory and the cardiac rhythms, and respiratory modulation of heart rate, are two competing aspects of cardio-respiratory interaction. From a physical viewpoint, synchronization and modulation are different phenomena and are related to different coupling. RSA generation is associated mainly with the baroreflex feedback loop and its respiratory phase-dependent information processing.” In our method, RSA has no role in the study of *BBI* vs. *mRRI* correlations and in the corresponding scatter plots, since we averaged all *RRI* values within each respiratory cycle, thereby annulling the intra-*BBI* heart rate variations and, consequently, the RSA effect, regardless of its amplitude. The results presented in [51], wherein the phase variability threshold for the automated synchrogram analysis of cardiorespiratory interactions in amateur cyclists was optimized, are interesting and offer a possible future sophistication of our method. Our results, presented in Figure 14a, wherein the standard deviation of *b1* is increased in orthostasis, are in accordance with the results of [51], where phase synchronization was reported to be reduced during standing, although the two results were obtained using very different approaches. Next, it would be interesting to combine the optimizing procedure described in [51] with our notion of *b1* synchronization, as a step toward the automated threshold-depended detection of *b1* synchronization. In addition, we could extend our method to study the temporal dependence of *b1*, an approach introduced and practiced in conventional synchrograms.

All the results presented in this paper obtained with healthy subjects, especially those referring to the positional parameters of the shared RR interval and the corresponding inspiration onset (Figure 12c,d), deserve evaluation in experimental studies with measures of vagal and sympathetic activity, continuous arterial blood pressure measurement, and measures of cardiac oxygen consumption. Along these lines, any condition altering the sympatho-vagal balance is of interest. The parameters are likely to be altered in physical exercise (treadmill, brisk walking, etc.), meditation, and sleep stages; these different pathological states all deserve extensive and elaborate investigation [39,52].

## Figures and Tables

**Figure 1 bioengineering-10-00180-f001:**
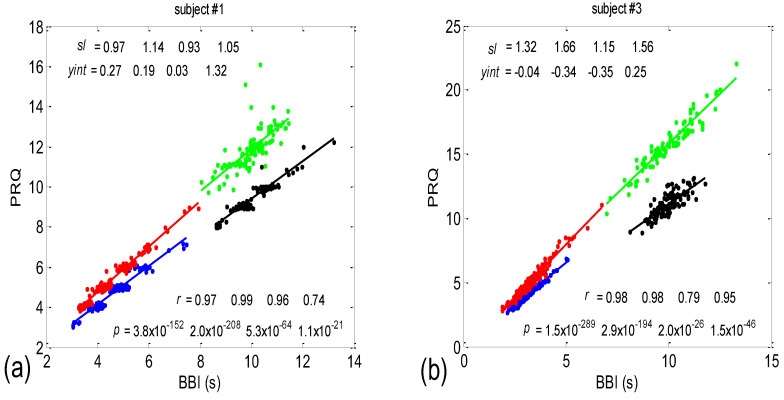
Two scatter plots of the *BBI* vs. *PRQ* correlations in two different subjects. (**a**) A case study with quantized *PRQ* values; (**b**) a case study with continual point distributions. Blue—supine; red—standing; black—supine, with slow 0.1 Hz breathing; green—standing with slow 0.1 Hz breathing. Linear regression slopes (*s_l_*); Y—intercepts (*yint*). Pearson’s coefficient of linear correlation (*r*) and regression significance (*p*) are also indicated for the four states, respectively, in the order indicated above.

**Figure 2 bioengineering-10-00180-f002:**
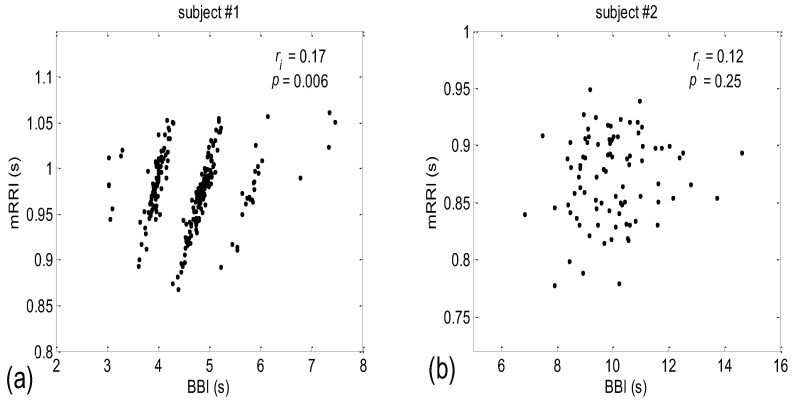
Examples of two types of scatter plots (*BBI* and *mRRI*) obtained from subjects #1 and #2 in supin and stand01 experimental conditions, respectively. (**a**) Subject #1 (supin), with pronounced quasi-linear clusters; (**b**) subject #2 (stand01), with no apparent clusters and with points randomly scattered. Upper right corner: *r_i_*—the integral (not cluster-averaged) coefficient of linear correlation is low and bears little physiological relevance; *p* is the corresponding significance level of *r_i_*.

**Figure 3 bioengineering-10-00180-f003:**
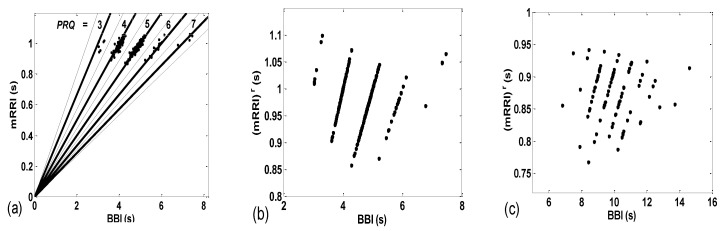
(**a**) The same data as seen in Figure 2a, with concurrent *PRQ_int_* lines added. Solid lines represent *PRQ_int_*; dashed lines are used for *PRQ_int_* ± 0.5, representing the cluster borders; (**b**,**c**) shows the same data as in Figure 2a,b, respectively, drawn with (*mRRI*)*^r^*, referring only to integer parts of *PRQ* values, according to Equation (3), and showing that all scattering information is thereby lost.

**Figure 4 bioengineering-10-00180-f004:**
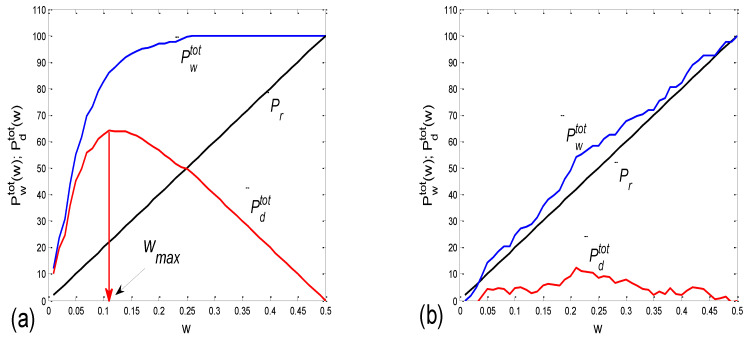
Example scatter plots: (**a**) with an aggregation of points around the concurrent *PRQ_int_* lines; (**b**) with a random distribution of points. *P_w_^tot^*—the percentage of points that lie within the gradually widening sectors, with a half-width of *w*, calculated for all clusters simultaneously, according to Equation (5); *P_r_*—the same quantity of uniformly distributed points; *P_d_^tot^*—their difference according to Equation (6); *w_max_*—the half-width where the maximum of *P_d_^tot^* occurs.

**Figure 5 bioengineering-10-00180-f005:**
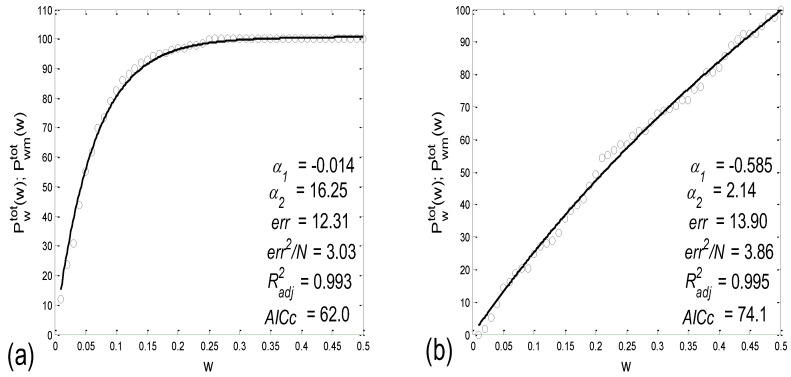
Examples of fitting the experimentally derived points *P_w_^tot^*(*w*) (open circles) with the model *P_wm_^tot^*(*w*) using Equation (9) (solid line). Four estimates for the quality of fitting are also given: error of fitting (*err*), square error per point (*err^2^*/*N*), R-square adjusted (*R^2^_adj_*), and Akaike’s information criterion (*AICc*). (**a**) An example showing pronounced aggregation, where *P_w_^tot^*(*w*) > *P_r_*; (**b**) another example close to a random distribution of points, where*P_w_^tot^*(*w*) ≈ *P_r_*.

**Figure 6 bioengineering-10-00180-f006:**
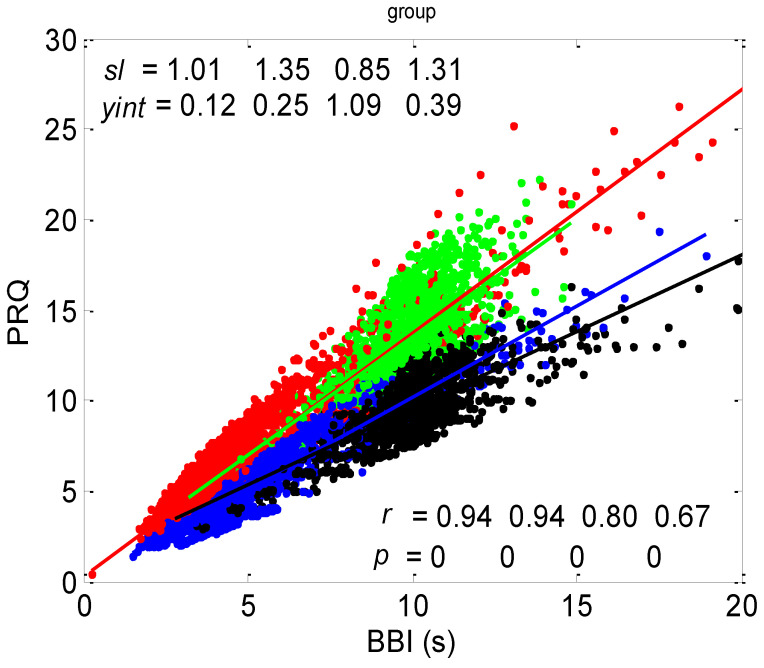
Diagram showing the group scatter plot of correlations between *BBI* and *PRQ*, where the results from twenty subjects in four states were superimposed. Each dot represents the result for one BB interval. The group linear regression for each state is drawn as a solid line: blue—supine; red—standing; black—supine, with slow 0.1 Hz breathing; green—standing, with slow 0.1 Hz breathing. Linear regression slopes (*sl*) with Y-intercepts (*yint*). Pearson’s coefficients of linear correlation (*r*) and regression significance (*p*, showing very high significance) are indicated as well, ordered from left to right as: supin, stand, supin01, and stand01.

**Figure 7 bioengineering-10-00180-f007:**
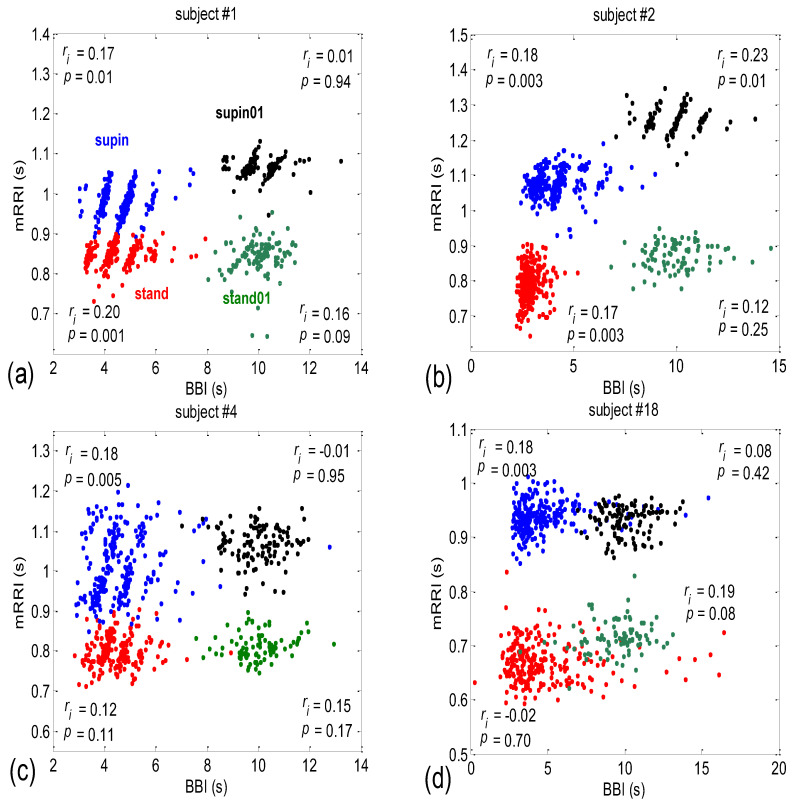
Four examples of subjects exhibiting different patterns of clustering, with high intra-cluster correlations of *BBI* vs. *mRRI* data. (**a**) Subjects where the scatter plots had clusters in all four experimental conditions; (**b**) subjects with clusters appearing only in the supine body position; (**c**) clusters appearing only in the case of spontaneous breathing; (**d**) subjects with no visible high-correlation clusters. The integral state-related Pearson’s coefficients (*r_i_*) and the corresponding correlation significances (*p*) are also shown. Each dot corresponds to one BB interval. Color coding is explained in (a).

**Figure 8 bioengineering-10-00180-f008:**
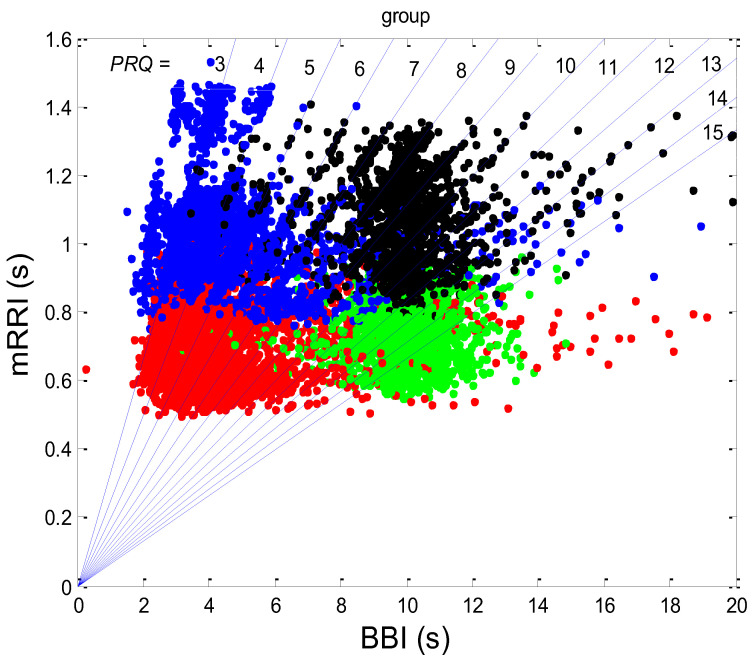
Group scatter plot of respiratory *BBI* vs. *mRRI* (averaged *RRI* within each *BBI*), obtained by superimposing separate scatter plots of all twenty subjects in all four states (supine—blue; standing—red; supine with slow 0.1 Hz breathing—black; standing with slow 0.1 Hz breathing—green). The concurrent *PRQ_int_* lines are also plotted.

**Figure 9 bioengineering-10-00180-f009:**
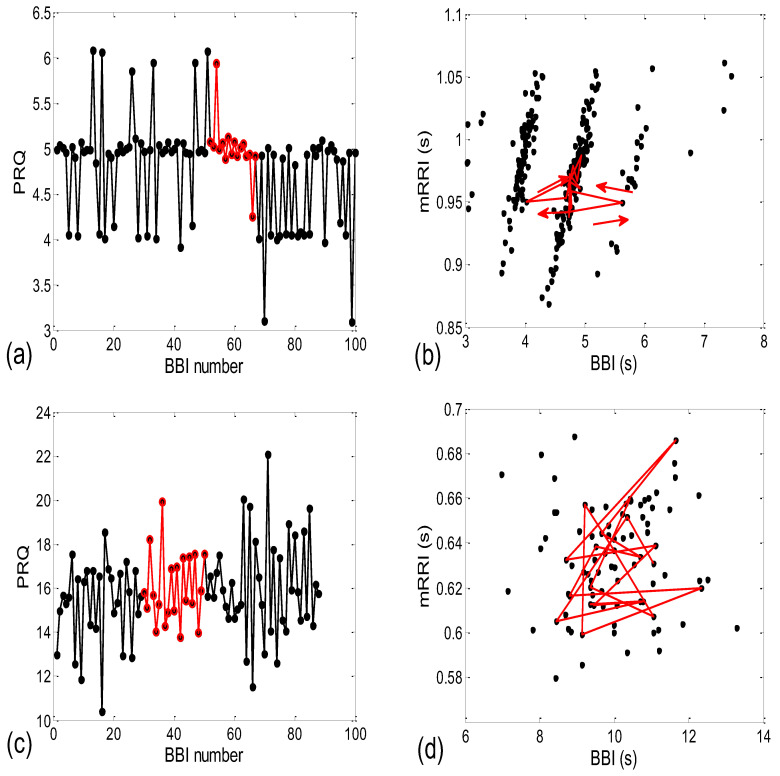
(**a**,**c**) The pulse–respiration quotient as a function of the *BBI* serial number (i.e., time, shown in black lines and black circles), with an arbitrarily chosen time interval colored in red; (**b**,**d**) the same data are presented as scatter plots of (*BBI* and *mRRI*) points. (**a**,**b**) Example of a *PRQ* time plot and the corresponding scatter plot, with pronounced clustering; (**c**,**d**) a *PRQ* time plot and its scatter plot, with a random distribution of points. The arrows in (**b**) indicate the directions of the point dynamics.

**Figure 11 bioengineering-10-00180-f011:**
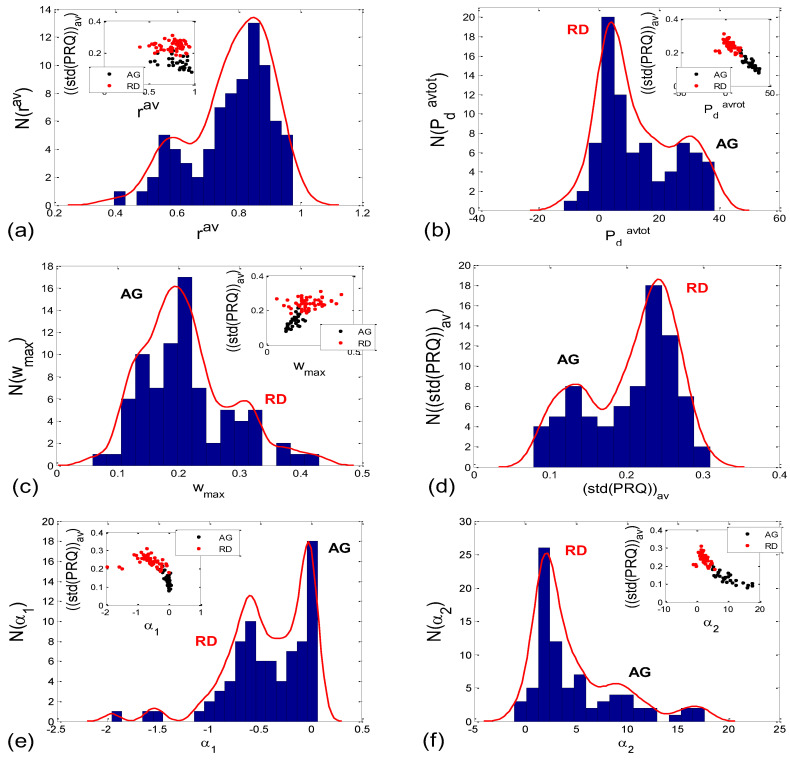
(**a**–**f**) Histograms approximating the distributions of 80 (20 subjects × 4 states) *BBI* vs. *mRRI* scatter plots over six proposed measures of aggregation of the points around the *PRQ_int_* lines. Corresponding PDE distributions are drawn as red solid curves. Except in the case of *r^av^*, all other five measures were pointing to two types of scatter plots (AG and RD), which is in accordance with the results in Figure 10 and Table 1 and can be assumed by the previous visual observation. Inlets show the correlations between a particular measure and (std(*PRQ*))_av_, where each dot represents one scatter plot, color-coded to correspond to one of two visually assessed subtypes (AG or RD).

**Figure 12 bioengineering-10-00180-f012:**
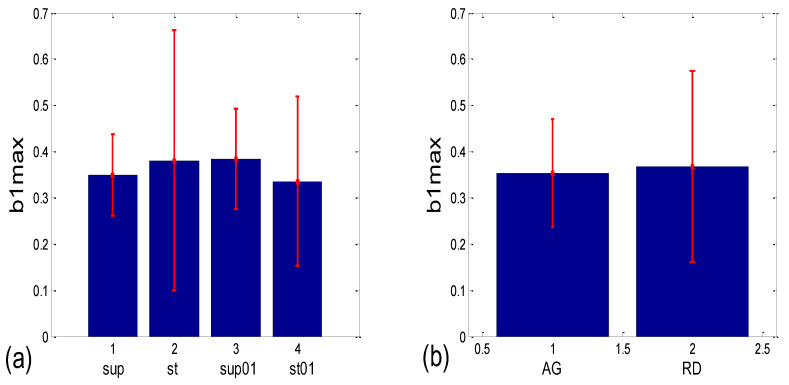
(**a**,**b**) Bar plots showing the results of the linear averaging of *b1max* (mean ± standard deviation); (**c**,**d**) (*b1a*)*_g_*—doubly angularly averaged from *b1*(*i*))_s_, according to Equations (10) and (11), across all *BBI*(*i*) and all subjects (*s*); the error bars represent the circularly corrected standard deviations (*b1_hst_*)*_c_*. (**a**,**c**) Statistics of the scatter plots in four experimental states; (**b**,**d**) the scatter plots shown according to the type of aggregation.

**Figure 13 bioengineering-10-00180-f013:**
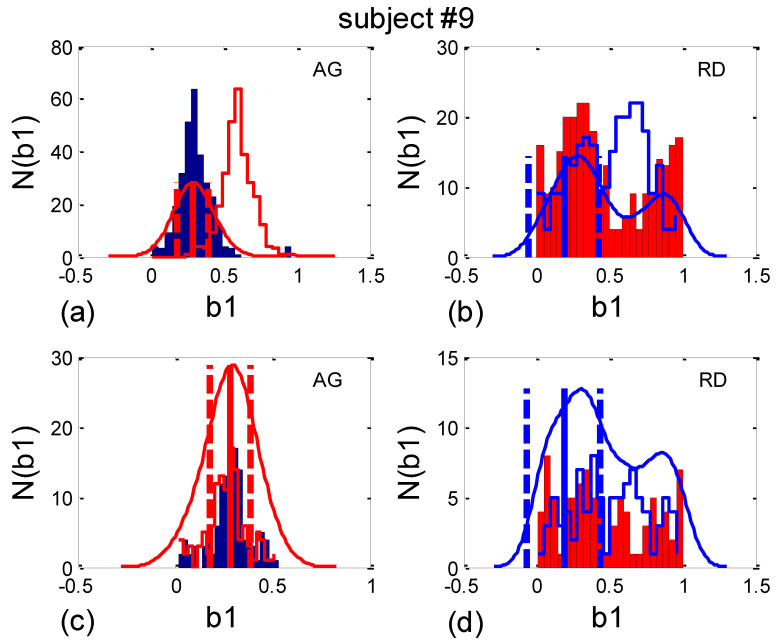
An example of the circular correction of the *b1* histogram-derived standard deviation, denoted in the text as (*b1_hst_*)*_c_*. (**a**) Supine; (**b**) standing; (**c**) supine with 0.1 Hz breathing; (**d**) standing with 0.1 Hz breathing. Their AG and RD codes are given in the right upper corner and the histograms are color-coded (AG—blue, RD—red). The PDE curves are drawn using opposite colors, for clarity. Centered histograms, having minimal standard deviation, are plotted as opposite-color open profiles. Angular mean values (vertical solid lines) are shown as ± circularly corrected standard deviation (vertical dashed lines).

**Figure 14 bioengineering-10-00180-f014:**
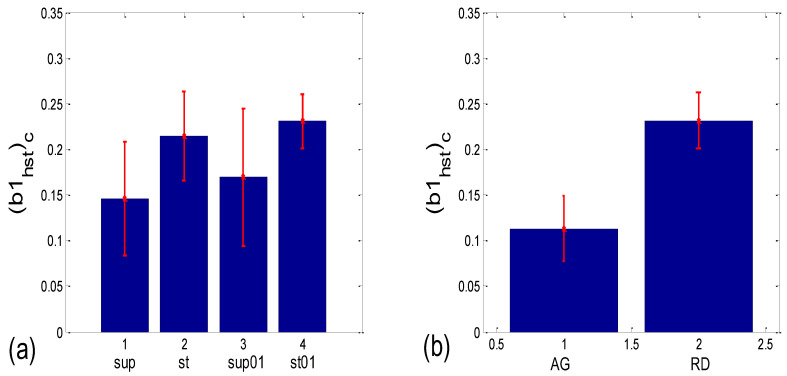
Group statistics of the circularly corrected histogram-derived standard deviation of *b1*, denoted as (*b1_hst_*)_c_, which is an inverse measure of *b1* locking. Comparison of: (**a**) experimental states; (**b**) scatter plots with aggregation and the random distribution of (*BBI* and *mRRI*) points (AG and RD, respectively).

## Data Availability

All the data are available for sharing with other investigators on the basis of the reciprocal exchange of data.

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
