# Peer review of "Two Operational Modes of Cardio-Respiratory Coupling Revealed by Pulse-Respiration Quotient"

_bioengineering, 2023, doi:10.3390/bioengineering10020180_

Round 1

Reviewer 1 Report

The authors utilize a logic based on heart period, measured as the RR interval from the electrocardiogram, to compute the pulse-respiration quotient (PRQ). This logic introduces the issue of R-wave peaks that do not belong to the considered breathing period. The authors address this issue and try to elucidate the relevance of accounting for border phenomena in the computation of PRQ based on RR interval via a linear regression analysis of PRQ and its portions on breathing period.

The contribution is interesting and rich of data and physiological considerations. However, some issues need clarification.

1)      It is expected that sympathetic activation induced by active standing reduces the strength of the causal dependence from respiration to RR interval series (see A. Porta et al, Comput Biol Med, 42, 298-305, 2012). It is unclear whether the present study confirms this result.

2)      Discussion should be enlarged by accounting for modifications of cardiorespiratory phase synchronization during active standing reported via synchrogram analysis (see B. Cairo et al, Phil Trans R Soc A, 379, 20200251, 2021). Cardiorespiratory phase synchronization is reduced by orthostatic challenge likely because it is a vagal phenomenon. Again, discussion should be enlarged.

3)      It remains unclear whether some terms of the decomposition of the PRQ into PRQint, b1 and b2 could be utilized to assess directionality of interactions. It is well-known that bidirectional interactions between respiratory system and the heart are present and can be quantified (see e.g., A. Porta et al, Phil Trans R Soc A, 371, 20120161, 2013). Discussion could be enlarged at this regard.

4)      The authors should elucidate how their conclusions change whether the number of cardiac beats per respiratory cycle is considered instead of the number of RR intervals per respiratory cycle.

5)      It remains unclear the final relevance of decomposing PQR in its portions especially because the use of the number of cardiac beats per respiratory cycle would not require any decomposition and it would be more linked to cardiac electrical activity.

6)      The authors should consider shortening and making more compact their contribution especially because some methodological considerations have been already published in a recent paper (see ref #16).

7)      Please check if the locution "breath interval increases" is always correct throughout the manuscript. It seems to be that sometimes it should be substituted with "breathing rate increases".

Reviewer 2 Report

In this paper, the pulse respiration quotient is further studied from three aspects: the relationship between BBI and PRQ, the relationship between BBI and the average RRI (mRRI) within each BBI, and the use of b1 to study the position of R pulse relative to the start of respiration. In this paper, a variety of data processing methods are used to realize the author's research, and the analysis is carried out in combination with two operating modes of cardiopulmonary coupling. However, in the process of reviewing the draft, several problems were also found that need to be pay more attention. Therefore, MAJOR revision has to be done before this manuscript could be accepted for publication in the MDPI.

Major comments:

The experiment in this paper includes four physiological states: spontaneous breathing (supin), standing with spontaneous breathing (stand), supine position with 0.1 Hz breathing (supin01) and standing with 0.1 Hz breathing (stand01). However, the setting of spontaneous breathing is not appropriate, and the spontaneous breathing is artificial participation, not free state, so I think it is meaningless. I hope the author can explain the purpose and significance of the experiment. It is suggested to add exercise conditions, such as five minutes on the treadmill or brisk walking, to enhance the significance of the experiment.

Minor comments:

1.   Line 281-randomly (uniformly),in my opinion, random and uniform here should not be confused. Pr here refers to evenly distributed points, which cannot be described by random. Therefore, I have doubts about the random and uniform here.

2.   Line 323,the formula for (std(PRQ))ic is not explained.

3.   Line 345,the original title of “2.7.1β1 locking” should be changed to “2.7.1β1 locking” because the location of the point is incorrect.

4.   Line 409,PR should be changed to PRQ.

5.   Line 445-460,is high intra-cluster correlations indicated by the author by visual correlations or by indicators? high intra-cluster correlations are not obvious in several scatter plots. Is it indicated by indicators? Please explain.

6.   Line 716 and 776,the titles "4.3. CR synchronization" and "4.3. Cardiorespiratory synchronization" have repetitive meanings.

Author Response

Rev 2

In this paper, the pulse respiration quotient is further studied from three aspects: the relationship between BBI and PRQ, the relationship between BBI and the average RRI (mRRI) within each BBI, and the use of b1 to study the position of R pulse relative to the start of respiration. In this paper, a variety of data processing methods are used to realize the author's research, and the analysis is carried out in combination with two operating modes of cardiopulmonary coupling. However, in the process of reviewing the draft, several problems were also found that need to be pay more attention. Therefore, MAJOR revision has to be done before this manuscript could be accepted for publication in the MDPI.

Major comments:

The experiment in this paper includes four physiological states: spontaneous breathing (supin), standing with spontaneous breathing (stand), supine position with 0.1 Hz breathing (supin01) and standing with 0.1 Hz breathing (stand01). However, the setting of spontaneous breathing is not appropriate, and the spontaneous breathing is artificial participation, not free state, so I think it is meaningless. I hope the author can explain the purpose and significance of the experiment. It is suggested to add exercise conditions, such as five minutes on the treadmill or brisk walking, to enhance the significance of the experiment.

Although more artificial than recording in a “freely moving” environment, recording spontaneous breathing in a laboratory setup was the only way that we could create the same experimental conditions as with slow paced 0.1 Hz breathing (as explained in section 2.2). We had a goal to compare physiological effects of spontaneous vs. slow breathing in two predefined body postures (supine vs. standing). By recording subjects out of the laboratory (walking in nature or moving in a home setting and using  some kind of halter equipment), we could not have been able to impose and control the subject to sit for 20 min, stand up, allow 5 min of adaptation and then continue standing for another 20 min.  Since we already published three papers reporting the results using such a setup, we thought that this should be a logical continuation of this line of research.

Thank you for the suggestion to add new exercise conditions to be analyzed in our research. It is an excellent idea, as it would generate new scatter-plots, where new distributions of points could be studied. As well, new data would be generated regarding the position of an average R impulse related to the inspiration onset. However, since the present work is already sufficiently long and contains enough results as it is, we are planning these new experiments to be performed in future. We added your suggestions in the Discussion, p. 26, lines 824-827.

Minor comments:

  1. Line 281-randomly (≈uniformly),in my opinion, random and uniform here should not be confused. Prhere refers to evenly distributed points, which cannot be described by random. Therefore, I have doubts about the random and uniform here.

You are right, we erased “randomly”. Initially, we had in mind that we deal with a “big enough randomly distributed number of points with a uniform probability distribution”, but it is more correct to say simply “uniform”.

  1. Line 323,the formula for (std(PRQ))ic is not explained.

On lines 321-322 we explained it with the following words:

“…standard deviation around PRQint of the corresponding intra-cluster set of data: (PRQ(i))ic = (BBI(i))ic/(mRRI(i))ic, i = 1, …, nic, denoted as (std(PRQ))ic,…”

However, to be clearer, we added these insertions:

“…conventional standard deviation, using MATLAB command “std”, around PRQint of the corresponding intra-cluster set of data: (PRQ(i))ic = (BBI(i))ic/(mRRI(i))ic, i = 1, …, nic, denoted as (std(PRQ))ic,…”

  1. Line 345,the original title of “2.7.1β1 locking” should be changed to “2.7.1β1 locking” because the location of the point is incorrect.

Thank you, our mistake, we corrected it (now line 344).

  1. Line 409,PR should be changed to PRQ.

Thank you again, our mistake, we added the missing “Q” (now line 408).

  1. Line 445-460,is high intra-cluster correlations indicated by the author by visual correlations or by indicators? high intra-cluster correlations are not obvious in several scatter plots. Is it indicated by indicators? Please explain.

In this initial stage it was done by visual observation. Since we were dealing with real biomedical data, there were some transition cases where intra-cluster correlations were higher than in others. In order to check which number of aggregation types we are dealing with, we proposed six quantitative aggregation measures and constructed their scatter-plot incidence histograms (Fig. 11). Our aim was to see whether histogram profiles (and the corresponding PDE curves) are, at least to a certain extent, bimodal. As expected, they were, although histogram profiles showed that the two peaks were not ideally separated. Inter-peak depletion zones were not detected, confirming your assessment that there were some variations in the intensity of aggregation. Further check of how correct was the initial visual identification, was obtained by constructing correlation diagrams between pairs of proposed measures, where each dot corresponds to one scatter-plot (inlets on panels of Fig. 11). Color coding of dots according to the initial visual identification (black – AG, red – RD), showed that red and black clouds were successfully separated in 2D, and that lack of inter-peak depletion zones was caused by projecting the 2D points on the particular x-axis (aggregation measure). The only exception was the histogram on Fig. 11a, where bimodality was not caused by the AG/RD separation, and which has yet to be investigated. In future, we plan to perform a further sophistication of AG/RD separation by applying some automated classification algorithms on pairs (2D) or triplets (3D) of aggregation measures.  

  1. Line 716 and 776,the titles "4.3. CR synchronization" and "4.3. Cardiorespiratory synchronization" have repetitive meanings.

Yes, we apologize; the subtitle, now on line 796, was an unwanted extra, and was removed. Thank you.

Reviewer 3 Report

I think that overall, the study sounds good; findings are novel, the conclusions are appropriate. The manuscript contains interesting information for the readers of Bioengineering.

Author Response

We thank the reviewer for his review of the manuscript and his positive comments.

Round 2

Reviewer 1 Report

The manuscript has been improved. The authors replied satisfactorily to all my issues and took into account the suggestions given. 

Reviewer 2 Report

The author has explained whether to add the experimental conditions of free movement, and put the suggestions into the future research plan. Besides, other problems have also been solved. After modification, I think this article can be published.